

# Construction and integrated analysis of crosstalking ceRNAs networks in laryngeal squamous cell carcinoma

Yuehui Liu and Fan Ye

Department of Otorhinolaryngology Head and Neck Surgery, The Second Affiliated Hospital of Nanchang University, Nanchang, Jiangxi Province, People's Republic of China

## ABSTRACT

**Background**. Laryngeal squamous cell carcinoma (LSCC) is one of the most common malignant tumours of the head and neck. Recent evidence has demonstrated that lncRNAs play important roles in tumour progression and could be used as biomarkers for early diagnosis, prognosis, and potential therapeutic targets. The "competitive endogenous RNA (ceRNA)" hypothesis states that lncRNAs competitively bind to miRNAs through their intramolecular miRNA reaction elements (MREs) to construct a wide range of ceRNA regulatory networks. This study aims to predict the role of ceRNA network in LSCC, for advancing the understanding of underlying mechanisms of tumorigenesis.

**Material and Methods**. In this study, the functions of lncRNAs as ceRNAs in LSCC and their prognostic significance were investigated via comprehensive integrated expression profiles data of lncRNAs, mRNAs, and miRNAs obtained from The Cancer Genome Atlas (TCGA). Protein–protein interaction, gene ontology, pathway, and Kaplan–Meier curves analysis were used to profile the expression and function of altered RNAs in LSCC.

**Results**. As a result, 889 lncRNAs, 55 miRNAs and 1946 mRNAs were found to be differentially expressed in LSCC. These altered mRNAs were mainly involved in extracellular matrix organization, calcium signaling, and metabolic pathways. To study the regulatory function of lncRNAs, an lncRNA-mediated ceRNA network was constructed. This ceRNA network included 61 lncRNAs, seven miRNAs and seven target mRNAs. Of these RNAs, lncRNAs (TSPEAR-AS, CASK-AS1, MIR137HG, PART1, LSAMP-AS1), miRNA (has-mir-210) and mRNAs (HOXC13, STC2, DIO1, FOXD4L1) had a significant effect on the prognosis of LSCC.

**Conclusion**. The results of this study broaden the understanding of the mechanisms by which lncRNAs are involved in tumorigenesis. Furthermore, five lncRNAs (TSPEAR-AS, CASK-AS1, MIR137HG, PART1, LSAMP-AS1) were identified as potential prognostic biomarkers and therapeutic targets for LSCC. These results provide a basis for further experimental and clinical research.

Corresponding author
Yuehui Liu, 499740265@qq.com

## INTRODUCTION

Laryngeal squamous cell carcinoma (LSCC) is considered to be one of the most common malignant tumors of the head and neck. It is estimated that 13,150 new cases of laryngeal cancer were diagnosed in the United States in 2018, of which about 3,710 patients died (*Siegel, Miller & Jemal, 2018*). Approximately 60% of the patients were diagnosed with advanced (stage III or IV) cancer (*Groome et al., 2003*). Although there has been great progress in the treatment of laryngeal cancer, the survival rate in the past few decades is still very low, and has been exhibiting a downward trend (*Steuer et al., 2017*; *Wan et al., 2010*). These reports highlight the necessity for more research and innovation in this area.

Long non-coding RNA (LncRNAs), composed of >200 nucleotides (nt) without protein-coding ability, have been found to function at epigenetic, transcriptional, post-transcriptional and translational levels (*Khorkova, Hsiao & Wahlestedt, 2015*). Recent evidence has demonstrated that lncRNAs play important roles in tumor progression and could serve as biomarkers for early diagnosis, prognosis and potential therapeutic targets of various cancers (*Gong et al., 2014*; *Parasramka et al., 2016*; *Rupaimoole et al., 2015*; *Yuan et al., 2014*). Although more than 50,000 lncRNA genes have been cloned and identified in the human genome, only a small part of their biological functions have been verified experimentally. Therefore, more efforts should be dedicated towards revealing how lncRNAs play multiple biological functions in malignant tumors. As described in the ''competitive endogenous RNA(ceRNA)'' hypothesis, lncRNAs competitively bind to miRNAs through their intramolecular miRNA reaction elements (MREs) to construct a wide range of ceRNA regulatory networks (*Salmena et al., 2011*). Specifically, lncRNAs acting as endogenous miRNA sponge competitively bind to a limited pool of miRNAs, affecting the inhibition of miRNAs on target genes. In fact, miRNA, a non-coding RNA of approximately 22 nucleotides in length, has been found to play an important role in tumorigenesis (*Berindan-Neagoe et al., 2014*; *Di Leva, Garofalo & Croce, 2014*; *McGuire, Brown & Kerin, 2015*; *Shin & Chu, 2014*). MiRNAs interact with Argonaute (AGO) and other proteins to form RNA-induced silencing complexes that bind to 3'UTRs of target RNAs to degrade target RNA or prevent translation (*Di Leva, Garofalo & Croce, 2014*). Studies have shown that one miRNA can inhibit hundreds of transcripts, while one lncRNA can inhibit multiple miRNAs (*Friedman et al., 2009*; *Salmena et al., 2011*). These data indicate that there is a large and complex ceRNA regulatory network in cells.

It is now well established from a variety of studies that competitive binding of lncRNA to miRNA plays an important role in the development of LSCC. For instance, the overexpression of RP11-169D4.1, which is a target of miR-205-5p, inhibits the proliferation, migration, and invasion of LSCC cell lines as well as promotes apoptosis (*Zhao et al., 2017*). NEAT1 is a novel target of miR-107 which can stimulate the invasion and metastasis of LSCC through regulating miR-107/CDK6 pathway (*Wang et al., 2016b*). AFAP1-AS1 increases RBPJ expression by negatively regulating miR-320a, and prevents drug resistance of LSCC (*Yuan et al., 2018*). Currently, ceRNA mechanism is the most interesting in lncRNA-mediated tumorigenesis. Therefore, this work aims to predict the role of ceRNA

network in LSCC, for advancing the understanding of the underlying mechanisms for tumorigenesis.

The Cancer Genome Atlas (TCGA), which provides normalized transcriptome profiling data, has increased understanding of the genetic basis of cancer. In this study, a comprehensive analysis of transcriptome data of LSCC from TCGA was performed. Significantly differentially expressed lncRNAs, miRNAs and mRNAs were identified in LSCC through comparing transcriptome data between tumor and normal tissues. Further, the biological functions of aberrantly expressed mRNAs and miRNA were explored via DAVID, KOBAS and Cytoscape plug-ins (ClueGO, CluePedia). A ceRNA network was then established based on the miRNA-binding site on both lncRNAs and mRNAs. To establish reliable biological markers for LSCC, survival analysis on RNA in ceRNA networks was finally performed through the corresponding clinical information from TCGA. The findings of this study provide new insight into the potential regulatory roles of the identified RNAs and how they affect LSCC pathogenesis.

## MATERIALS & METHODS

### Study population

In this study, a comprehensive analysis of RNA-Seq data and clinical information of LSCC from the TCGA database was conducted. To realize this goal, a comprehensive search was firstly conducted in TCGA based on the following criteria: (Disease Type is Squamous Cell Neoplasms), (Primary Site is Larynx), (Program Name is TCGA), (Workflow Type is HTSeq-Counts), (Data Category is Transcriptome Profiling) and (Data Type is Gene Expression Quantification). The RNA-Seq data was then downloaded through a data transfer tool provided by TCGA. Finally, the RNA-Seq data of 111 tumor samples and 12 normal samples, and their corresponding clinical information was obtained. In addition, miRNA-Seq data specially used for analyzing differentially expressed miRNAs was downloaded. These miRNA-Seq data were obtained from 117 tumor samples and 12 normal samples. Detailed characteristics of the patients are shown in Table 1.

### Differentially expressed analysis of RNAs

The RNA-seq data was downloaded, after which the expression analysis of lncRNA, mRNA, and miRNA was performed. Based on previously reported methods (*Huang et al., 2017*), the different expression of RNAs between larynx cancer and normal tissues was evaluated using the "edgeR" package in R software with thresholds of $|\log 2 \text{foldChange(FC)}| > 2.0$ and adjusted $P$-value $< 0.01$. Volcano maps were then drawn to generate a graphical overview of their expression profile via "gplots" package in R software.

### Construct the ceRNA network

In the current study, the interaction of lncRNA-miRNA-mRNA was predicted in LSCC based on the overlapping of the miRNA seed sequence binding site on both lncRNAs and mRNAs. The interaction between abnormally expressed lncRNAs and miRNAs was firstly predicted through Mircode database which provides "whole transcriptome" human miRNA target predictions based on the comprehensive GENCODE gene annotation and

**Table 1 Characteristics of patients.**

| Characteristic | | Number |
|---|---|---|
| **Patient sex** | | |
| | Male | 97 (82.9%) |
| | Female | 20 (17.1%) |
| **Patient age (y) ($\bar{x} \pm$ SD)** | | 61.9 ± 9.1 |
| **Tumor stage** | | |
| | Stage I | 2 (1.7% ) |
| | Stage II | 10 (8.5% ) |
| | Stage III | 14 (12.0%) |
| | Stage IV | 74 (63.2%) |
| | unknown | 17 (14.6%) |
| **T category** | | |
| | T1 | 7 (6.0%) |
| | T2 | 14 (12.0%) |
| | T3 | 26 (22.2%) |
| | T4 | 55 (27.0%) |
| | unknown | 15 (12.8%) |
| **N category** | | |
| | N0 | 41 (35.0%) |
| | N1 | 12 (10.3%) |
| | N2 | 41 (35.0%) |
| | N3 | 2 (1.7%) |
| | unknown | 21 (18.0%) |
| **Vital status** | | |
| | alive | 74 (63.2%) |
| | dead | 43 (36.8%) |

includes >10,000 long non-coding RNA genes. The miRTarBase, TargetScan, and miRDB databases were then used to identify aberrantly expressed miRNA-mRNA pairs. This study included the target mRNAs which matched the databases only. Finally, lncRNA-miRNA and miRNA-mRNA pairs were merged into a ceRNA network based on their shared miRNAs. The Cytoscape v3.6.1 software was used to visualize the co-expression network of differentially expressed mRNAs, lncRNAs and miRNAs.

## Protein–protein interaction analysis

To study the protein–protein interaction (PPI) in LSCC, String (Search Tool for the Retrieval of Interacting Genes) online tools were used to construct a PPI network for significantly aberrant mRNAs. In brief, the list of top 500 abnormally expressed mRNAs were uploaded to String for analysis. The minimum required interaction score was set at 0.7, meaning that the association between proteins had high confidence. Cytoscape plug-in CentiScape was then used to analyze the PPI network and screen out its hub proteins. Finally, Cytoscape software was used to visualize the PPI networks.

## Functional & pathway enrichment analyses

Significantly aberrantly expressed mRNAs were used to conduct gene ontology (GO) and Kyoto Encyclopedia of Genes and Genomes (KEGG) pathway analysis via DAVID 6.8 and KOBAS 3.0 respectively. GO and pathway annotation networks was then established using Cytoscape plug-in ClueGO as described by *Bindea et al. (2009)*. The primary parameters used were as follows: showing only pathways with $p < 0.05$; GO term or pathway network connectivity ($\kappa$-score) $= 0.4$. In order to identify the function of differentially expressed miRNAs in laryngeal cancer, GO and KEGG enrichment were indirectly performed through their target genes. To this end, the target mRNAs were firstly predicted using CluePedia, a Cytoscape plug-in. Using ClueGO, bio-process and pathway analysis of these target mRNAs were then performed. To visualize the function of these miRNAs, miRNA-mRNA pairs and functional items were correlated and presented through Cytoscape.

## Survival analysis

To analyze the prognostic significance of differentially expressed RNAs in ceRNA network, survival analysis was conducted using clinical information from TCGA. As previously described (*Huang et al., 2017*), this study was based on Kaplan–Meier curve analysis. The survival curve was plotted using the "survival" package in R with the cutoff of $p < 0.05$.

# RESULTS

## Differentially expressed lncRNAs and mRNAs in LSCC

Gene expression quantification data of 111 cancer samples and 12 normal samples were downloaded via a data transfer tool provided by TCGA. lncRNAs and mRNAs were then extracted from the gene expression quantification data using computer script for differential analysis. Consequently, 889 lncRNAs and 1946 mRNAs were found to be aberrantly expressed, through the "edgeR" package in R software with thresholds of |log2foldChange (FC)|>2.0 and adjusted *P*-value <0.01. Of these RNAs, 617 lncRNAs and 1008 mRNAs were upregulated, while 272 lncRNAs and 936 mRNAs were downregulated. The top 30 of aberrantly expressed lncRNAs and mRNAs were presented in tables ranked in order of |log2foldChange (FC)| (Tables 2 and 3). To visually describe these differentially expressed lncRNAs and mRNAs between LSCC and normal group, their expression profiles were presented using volcano maps (Figs. 1A and 1B). The "gplots" package in R software was used to build volcano maps, which are based on adjusted *p*-value and fold change. The results suggested that LSCC was associated with not only the aberrantly expressed mRNAs but also with the altered lncRNAs.

## Protein–protein interaction & functional analyses of mRNAs

The top 500 differentially expressed mRNAs were input into PPI for analyses via the String online tools (minimum required interaction score = 0.7). To screen the hub protein in the PPI network, CentiScape was used to analyze the PPI network. A protein with connectivity of >10 as was defined as a hub protein based on the distribution density of nodes in the network. This resulted in 35 hub proteins which might play important roles in tumorigenesis of LSCC (Fig. 1C). The most significant hub proteins were actinin
**Table 2  The top 30 differentially expressed lncRNAs.**

| lncRNA | logFC | Adjusted P-value | Stage | lncRNA | logFC | Adjusted P-value | Stage |
|---|---|---|---|---|---|---|---|
| AC105460.1 | 9.624232519 | 3.47E-07 | up | LINC02303 | −8.124203643 | 4.01E-44 | down |
| AC100801.1 | 8.238931778 | 1.93E-05 | up | LINC00314 | −6.70567849 | 6.54E-22 | down |
| AC010595.1 | 8.138849584 | 2.31E-14 | up | AL137246.2 | −6.665159808 | 1.29E-24 | down |
| AC105460.2 | 7.716181392 | 8.04E-07 | up | SLC8A1-AS1 | −6.414996281 | 5.19E-106 | down |
| MAGEA4-AS1 | 7.494825221 | 0.0001893 | up | HCG22 | −6.335918315 | 3.33E-43 | down |
| AC073365.1 | 7.389003818 | 4.59E-05 | up | LINC02487 | −5.798837862 | 4.01E-32 | down |
| G2E3-AS1 | 7.222069873 | 1.00E-12 | up | MIR133A1HG | −5.601245414 | 1.11E-17 | down |
| LINC02582 | 7.138114151 | 0.000685436 | up | LINC02538 | −5.578220186 | 5.78E-35 | down |
| LINC02525 | 7.116487546 | 0.004481251 | up | LINC00330 | −5.497294479 | 2.86E-32 | down |
| AC079062.1 | 7.047002194 | 0.001397173 | up | LINC00443 | −5.31185955 | 5.51E-26 | down |
| AC087612.1 | 6.661860103 | 4.60E-07 | up | LINC01028 | −5.23325995 | 4.37E-18 | down |
| LINC01194 | 6.617984821 | 0.005956076 | up | AC103563.2 | −5.187886248 | 2.52E-19 | down |
| AC022639.1 | 6.591725463 | 1.60E-05 | up | AC005532.1 | −5.074133501 | 1.77E-39 | down |
| AFAP1-AS1 | 6.563017491 | 1.17E-06 | up | IL12A-AS1 | −5.062202771 | 1.56E-75 | down |
| AC092916.1 | 6.524150544 | 0.008798004 | up | LINC01765 | −5.013858116 | 1.85E-13 | down |

**Notes.**

FC,  fold change.

**Table 3  The top 30 differentially expressed mRNAs.**

| mRNA | logFC | Adjusted P-value | Stage | mRNA | logFC | Adjusted P-value | Stage |
|---|---|---|---|---|---|---|---|
| SOHLH1 | 10.25241617 | 1.23E-05 | up | PRB4 | −9.884354737 | 4.80E-30 | down |
| MAGEA12 | 9.968586173 | 0.000146917 | up | SCGB3A2 | −7.65197743 | 7.17E-44 | down |
| MAGEA1 | 9.639662076 | 0.000247667 | up | PRB1 | −7.186135395 | 3.77E-20 | down |
| MAGEC2 | 9.630175143 | 0.001526406 | up | MYH7 | −6.928426658 | 5.25E-21 | down |
| PAGE2 | 9.467661276 | 0.005362649 | up | FBXO40 | −6.7579754 | 1.95E-29 | down |
| PRR20G | 8.976961309 | 5.75E-06 | up | PYGM | −6.716881524 | 6.54E-36 | down |
| CTAG2 | 8.922913917 | 0.006038851 | up | KRT4 | −6.488075449 | 7.40E-30 | down |
| MAGEA3 | 8.79801806 | 6.40E-06 | up | DWORF | −6.411014171 | 7.40E-16 | down |
| ADAMTS20 | 8.728369712 | 3.35E-17 | up | CRNN | −6.407039436 | 2.36E-25 | down |
| LCE2B | 8.695652526 | 0.000497367 | up | DHRS7C | −6.382622791 | 7.23E-23 | down |
| SAGE1 | 8.660902318 | 0.000329428 | up | PIP | −6.36304819 | 1.53E-17 | down |
| MAGEA6 | 8.385660885 | 2.17E-05 | up | MYL3 | −6.334607727 | 2.82E-29 | down |
| FGF19 | 8.234213225 | 9.44E-05 | up | SMCO1 | −6.29838908 | 7.17E-51 | down |
| NOBOX | 8.171244913 | 0.002444398 | up | ASB11 | −6.111357822 | 3.61E-20 | down |
| CXorf67 | 8.083518872 | 0.005239453 | up | PRB3 | −6.04890691 | 2.37E-19 | down |

**Notes.**

FC,  fold change.

alpha 2(ACTN2), myosin heavy chain 6(MYH6), titin-cap (TCAP), troponin i2 (TNNI2), Late Cornified Envelope Protein (LCE), Small Proline-Rich Protein 2G (SPRR2G), Matrix Metallopeptidase 9 (MMP9) and Fibrinogen Alpha Chain (FGA).

The potential regulatory roles of the identified mRNAs and how they affect LSCC pathogenesis were then studied. Both GO and KEGG enrichment analyses were used to

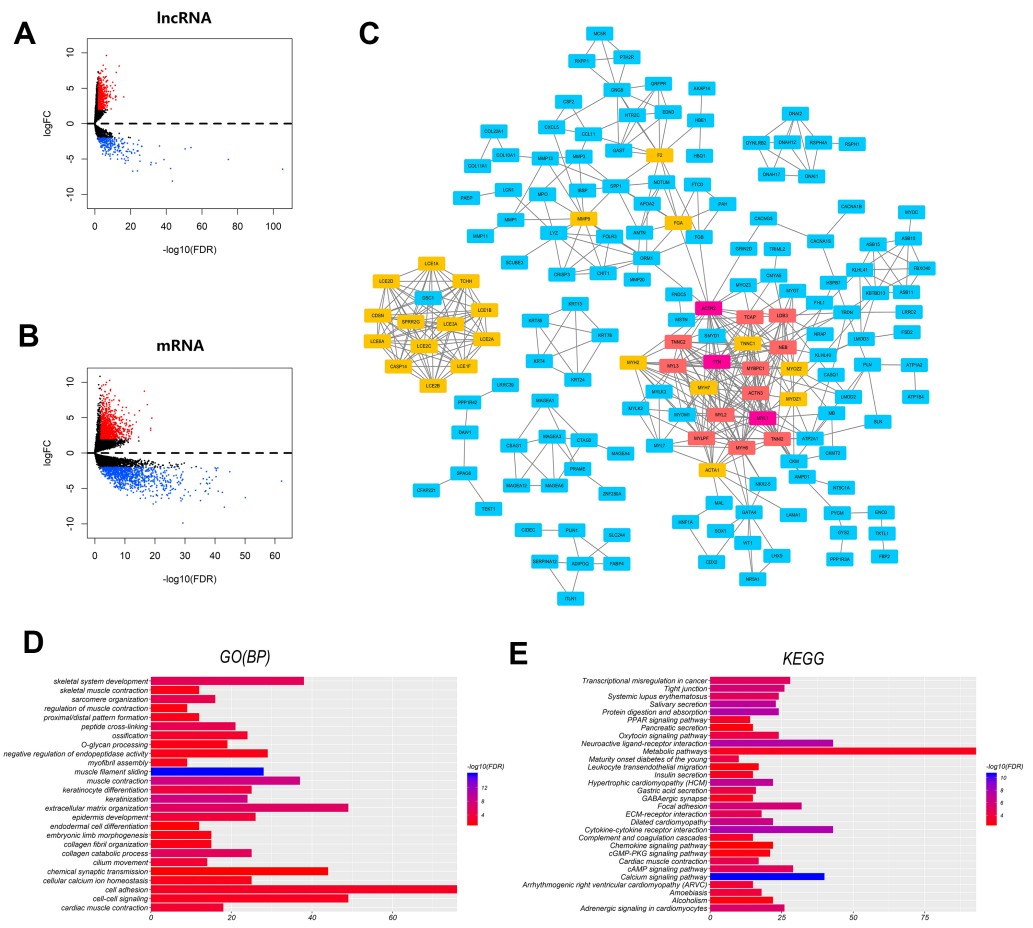

**Figure 1** **Volcano plots, protein-protein interaction, GO (BP) and KEGG enrichment analyses for differentially expressed mRNAs and long noncoding RNAs.** Volcano plots are used to visualize and assess the variation of (A) lncRNAs, (B) mRNAs expression between LSCC tissues and normal tissues. (C) The PPI networks of altered mRNAs. Each rectangle corresponds to a protein-coding gene (mRNA). Colors represent different connectivity (blue < 10, yellow ≥ 10, orange ≥ 15 and purple ≥ 20). Top 30 (D) biological processes and (E) pathways of significantly differentially expressed mRNAs in the GO analysis and KEGG analysis, respectively. The horizontal axis represents the number of enriched genes and the intensity of color represents corrected *P*-value.

determine the functions of aberrantly expressed mRNAs (see 'Materials & Methods'). GO enrichment analysis with DAVID indicated that differentially expressed mRNAs were mainly enriched in the following biological processes (BP): extracellular matrix organization, collagen catabolic process, skeletal system development, epidermis development, adhesion and cell–cell signaling (Fig. 1D). KEGG enrichment analysis via KOBAS showed that differentially expressed mRNAs were chiefly involved in the metabolic pathways, calcium signaling pathway, cytokine-cytokine receptor interaction, protein digestion and absorption, neuroactive ligand–receptor interaction and salivary secretion (Fig. 1E). In addition, some common tumor-related pathways, such as cAMP, Jak, PI3K signaling pathway and pathways in cancer (Fig. 1E) were also significantly

enriched. Cytoscape plug-in was then applied for Go and KEGG enrichment analysis to further verify biological processes and signaling pathways. GO enrichment analysis with ClueGO revealed that these mRNAs were involved in tissue development, multicellular organism development, muscle filament sliding, epithelium development, muscle system process, extracellular matrix organization, cell differentiation and extracellular structure organization (Figs. 2A–2K). Interestingly, the results of KEGG enrichment from ClueGO were very similar to those of KOBAS. This results indicated that differentially expressed mRNAs were involved in cytokine-cytokine receptor interaction, maturity onset diabetes of the young, salivary secretion, calcium signaling pathway, protein digestion and absorption, neuroactive ligand–receptor interaction and ECM-receptor interaction (Figs. 2L–2O). Finally, the network of biological processes and signaling pathways was established base on co-expressed genes (Fig. 2). This network suggested that these bioprocesses and signaling pathways might be related to the pathogenesis and development of LSCC.

## Expression and functional analysis of altered miRNAs in LSCC

Differentially expressed lncRNAs and miRNAs were identified in laryngeal carcinoma. However, there was also a need to identify the expression of miRNAs for construction of ceRNA network. For this purpose, the miRNA expression quantification data of 117 LSCC samples and 12 normal samples were downloaded from TCGA. By comparing the LSCC group with the normal group, the significantly differentially expressed miRNAs were identified via the "edgeR" package in R software. By setting thresholds of $|log2foldchange(FC)|>2.0$ and adjusted $P$-value $<0.01$, 55 aberrantly expressed miRNAs in LSCC were revealed. Of these, 33 miRNAs were upregulated and 22 miRNAs were downregulated. The top 30 of these miRNAs are presented in Table 4. Profiling and functional analyses was then performed. Similarly, a graphical overview of the miRNAs expression profile was generated using a volcano plot (Fig. 3A). The volcano plot showed that the proportion of dysregulated miRNAs in laryngeal carcinoma is similar to lncRNAs or mRNAs. These results suggested that the lncRNA-miRNA-mRNA network might be an important regulatory mechanism in laryngeal cancer.

To study the function of aberrantly expressed miRNAs, their target mRNAs were firstly predicted through miRTarBase and Miranda databases. ClueGO was then used to conduct GO and KEGG enrichment analysis of these target mRNAs. Finally, miRNA-mRNA pairs were associated with biological processes and signaling pathways through CluePedia. The results revealed that EGFR, CTGF, COL4A2, FSTL1, CREB1, CDK4 and CDKN1A could be the key genes by which these miRNAs exerted regulatory function in LSCC (Fig. 3B). GO enrichment analysis indicated that these target genes were enriched in bioprocesses, such as positive regulation of fibroblast proliferation, cellular response to amino acid stimulus, cellular response to fatty acid and response to amino acid (Fig. 3B). Most importantly, KEGG analysis revealed that these target genes were involved in tumor-associated pathways, such as colorectal cancer, bladder cancer, non-small cell lung cancer, pancreatic cancer, melanoma and adherens junction (Fig. 3B). These results suggests that the potential regulatory roles of aberrantly expressed miRNAs in LSCC might be one of the mechanisms of LSCC tumorigenesis.

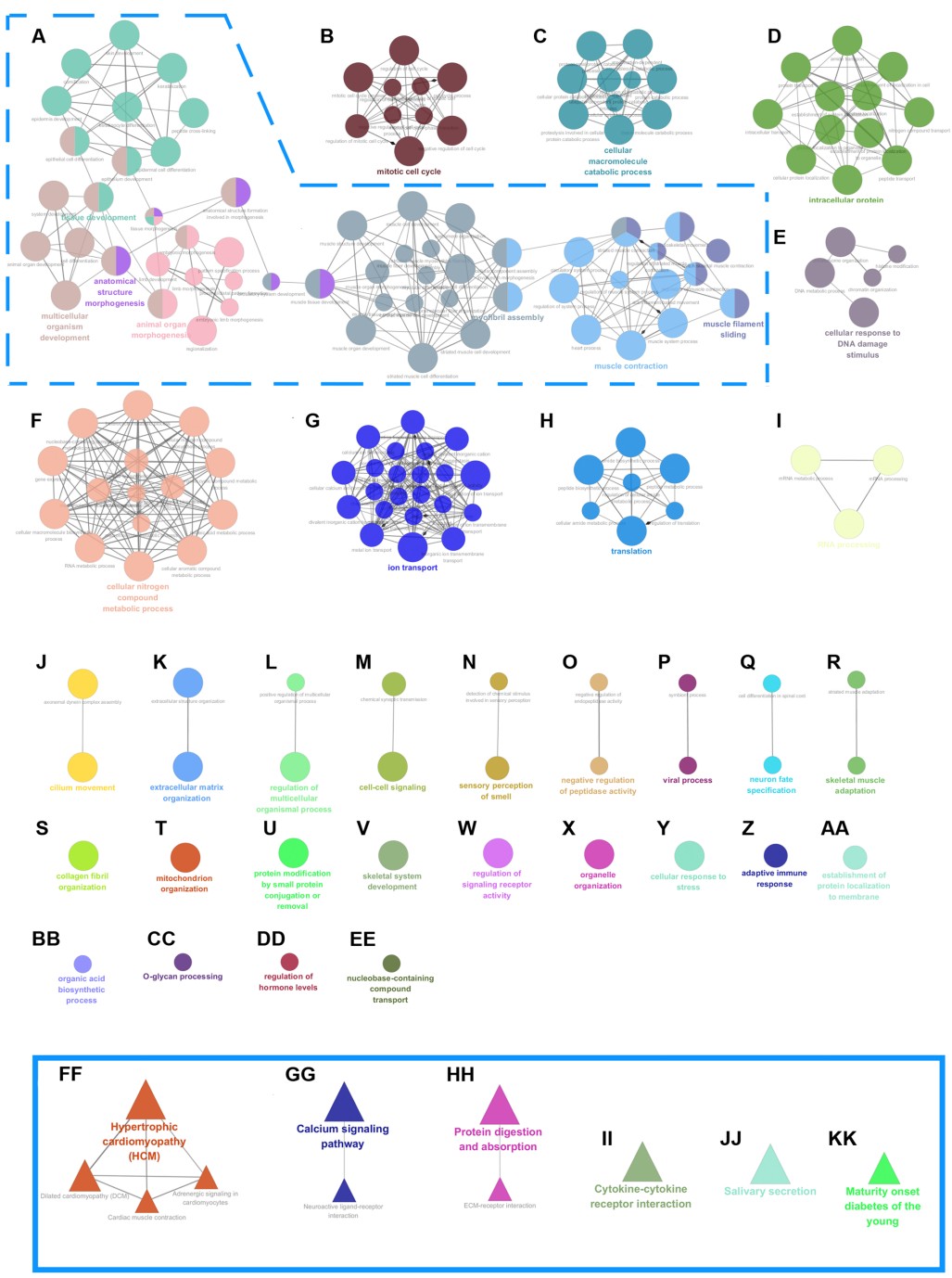

**Figure 2  Network view of processes and pathways of aberrantly expressed mRNAs using ClueGO and CluePedia.** Functionally grouped network with (A-EE) processes and (FF-KK) pathways as nodes linked based on their co-expressed genes ($\kappa$-score $\geq$ 0.4). The most significant processes and pathways in each group are marked. The node size represents enrichment level.

**Table 4    The top 30 differentially expressed miRNAs.**

| miRNA | logFC | Adjusted *P*-value | Stage | miRNA | logFC | Adjusted *P*-value | Stage |
|---|---|---|---|---|---|---|---|
| hsa-mir-105-2 | 6.46 | 0.000101469 | up | hsa-mir-378i | −4.98512 | 1.03E-20 | down |
| hsa-mir-105-1 | 6.23 | 8.60E-05 | up | hsa-mir-1-1 | −4.12954 | 1.26E-09 | down |
| hsa-mir-767 | 6.17 | 4.09E-05 | up | hsa-mir-208b | −4.1141 | 6.31E-07 | down |
| hsa-mir-1269b | 5.73 | 0.009537851 | up | hsa-mir-1-2 | −4.10462 | 1.68E-09 | down |
| hsa-mir-615 | 5.32 | 2.27E-10 | up | hsa-mir-133b | −3.90956 | 1.42E-07 | down |
| hsa-mir-196a-2 | 4.87 | 9.52E-09 | up | hsa-mir-133a-1 | −3.64194 | 1.42E-07 | down |
| hsa-mir-196a-1 | 4.73 | 3.77E-08 | up | hsa-mir-133a-2 | −3.55484 | 2.29E-07 | down |
| hsa-mir-514a-3 | 4.53 | 0.006426286 | up | hsa-mir-449b | −3.47533 | 1.10E-06 | down |
| hsa-mir-548f-1 | 4.53 | 0.00080213 | up | hsa-mir-375 | −3.36809 | 9.26E-13 | down |
| hsa-mir-514a-1 | 4.35 | 0.008651113 | up | hsa-mir-449a | −3.22293 | 5.14E-08 | down |
| hsa-mir-9-3 | 4.10 | 2.62E-05 | up | hsa-mir-378d-1 | −2.83308 | 5.25E-13 | down |
| hsa-mir-9-2 | 4.07 | 2.71E-05 | up | hsa-mir-378d-2 | −2.69707 | 3.01E-11 | down |
| hsa-mir-9-1 | 4.07 | 2.71E-05 | up | hsa-mir-499a | −2.57694 | 0.000641862 | down |
| hsa-mir-1269a | 4.01 | 0.00052332 | up | hsa-mir-206 | −2.44746 | 0.003311076 | down |
| hsa-mir-1910 | 3.85 | 6.34E-09 | up | hsa-mir-6510 | −2.39858 | 6.35E-08 | down |

Notes.

FC, fold change.

## LncRNA-mediated ceRNA network revealed potential mechanisms of LSCC tumorigenesis

To explore the regulatory mechanism between the lncRNA and mRNA transcripts, the lncRNA–miRNA–mRNA network was established based on the ceRNA hypothesis via their integrating expression profile data and their regulatory relationships. As described in 'Materials & Methods', the interaction between differentially expressed lncRNAs and miRNAs were firstly identified. The results revealed 114 lncRNA-miRNA pairs, consisting of 61 LncRNAs and seven mRNAs (Table 5). MiRTarBase, TargetScan and miRDB database were then used to predict miRNA-mRNA pairs. All the target mRNAs which matched with the databases were included in this study. A total of 90 target mRNAs (Table 6) were found. However, not all of these predicted target mRNAs were differentially expressed in LSCC. Therefore, the intersection of predicted mRNAs and all differentially expressed mRNAs was taken in the current study (Fig. 3C). Finally, a co-expression network of RNAs was built by merging lncRNA-miRNA pairs and miRNA-mRNA pairs based on their shared miRNAs. As a result, 61 lncRNAs, seven miRNAs and seven target mRNAs were included in the ceRNA network (Table 5), and their regulatory relationships were visualized by Cytoscape (Fig. 3D). This ceRNA network showed that complex interactions of lncRNA-miRNA-mRNA might be a potential cause of gene expression disorders in LSCC. GO and KEGG enrichment analysis showed that the target genes of ceRNA were mainly enriched in the following biological functions and pathways: biosynthetic process, regulation of macromolecule metabolic process, regulation of cellular metabolic process, hormone biosynthetic process, glycerophospholipid metabolism, thyroid hormone signaling pathway and transcriptional misregulation in cancer (Figs. 4A and 4B). These results suggest that lncRNA-mediated ceRNA network might play an important role in tumorigenesis.

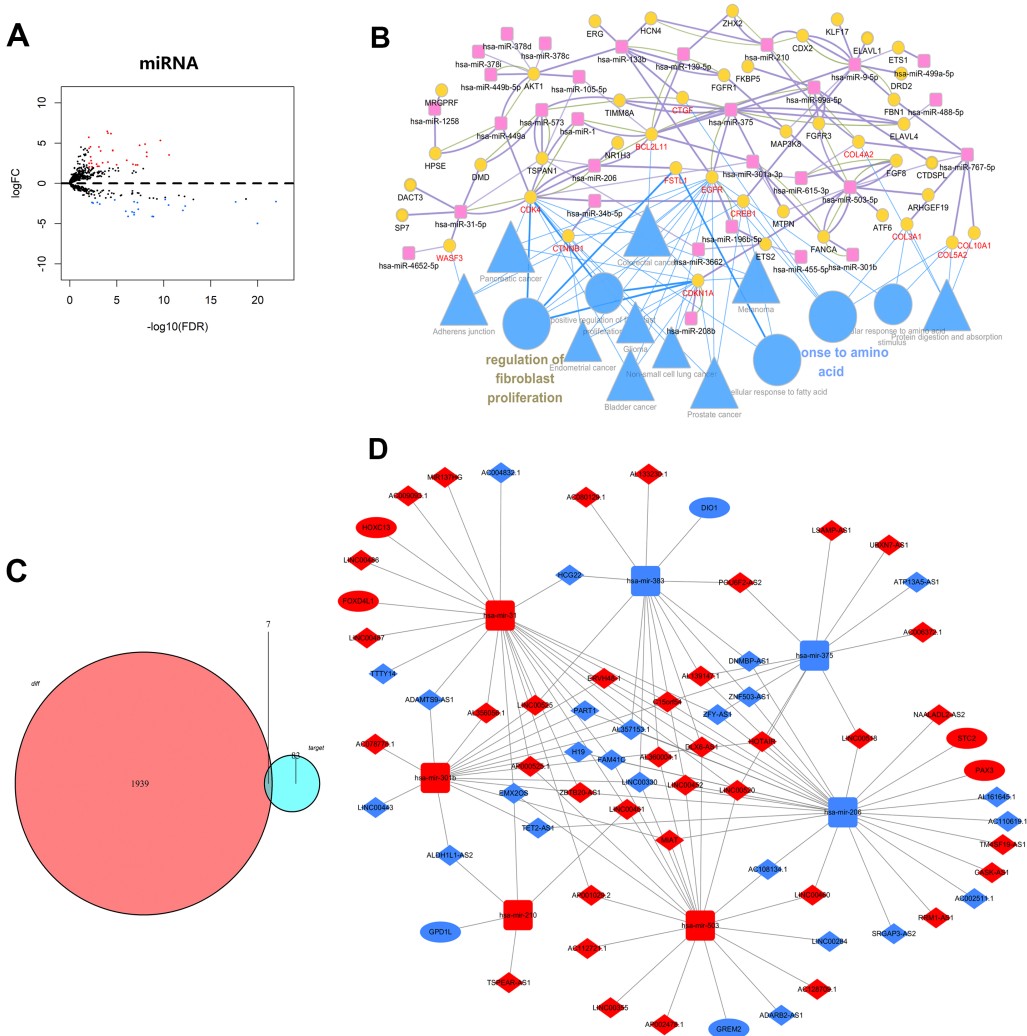

**Figure 3  Functional analysis of differentially expressed miRNA and ceRNA network in LSCC.** (A) Vocalno plot indicating the significantly differentially expressed miRNAs in LSCC ($p < 0.01$ and fold change of $> 2$). (B) GO and KEGG enrichment analysis of the target genes of altered miRNAs. The association between miRNA-mRNA pairs, biological processes, and signaling pathways was determined by CluePedia. Pink squares represent differentially expressed miRNAs, and yellow circles represent target genes. Blue circles and triangles represent processes and pathways, respectively. (C) Venn diagram showing the intersection of the predicted mRNAs and all differentially expressed mRNAs. (D) The global view of the ceRNA network in LSCC. Diamonds and squares represent lncRNAs and miRNAs, respectively. Ovals represent mRNAs. Red nodes represent upregulation and blue nodes represent downregulation.

Survival analysis was performed to study the prognostic significance of ceRNA network. The expression profile data of RNAs in the ceRNA network and clinical information from TCGA was integrated to establish Kaplan–Meier curves. The results revealed that the aberrant expression of lncRNAs (TSPEAR-AS, CASK-AS1, MIR137HG, PART1, LSAMP-AS1), miRNA (has-mir-210) and mRNAs (HOXC13, STC2, DIO1, FOXD4L1) had significant effect on the prognosis of LSCC (Figs. 4C–4L). The results suggest that

**Table 5  The differentially expressed lncRNAs, miRNAs, and mRNAs included in ceRNA network.**

| The type of RNAs | Gene symbols |
|---|---|
| LncRNA | H19,LINC00525,PART1,C15orf54,AL357153.1,TTTY14, AP002478.1,LINC00518,AC009093.1,AL360004.1,AC004832.1, ADARB2-AS1,AC002511.1,LINC00487,AC108134.1, AP000525.1,MIAT,UBXN7-AS1,ZNF503-AS1,NAALADL2-AS2,LINC00355,DNMBP-AS1,HOTAIR,SRGAP3-AS2,HCG22, LINC00452,EMX2OS,LINC00443,FAM41C,AL161645.1, LINC00486,MIR137HG,DLX6-AS1,CASK-AS1,ERVH48-1,ZFY-AS1,LINC00460,LINC00284,POU6F2-AS2,LINC00330,TSPEAR-AS1,TM4SF19-AS1,ATP13A5-AS1,AC006372.1,AC112721.1, AL133230.1, AC080129.1,AL356056.1,LSAMP-AS1, ADAMTS9-AS1,ZBTB20-AS1, AC078778.1,LINC00461,ALDH1L1-AS2, AL139147.1,AC128709.1, TET2-AS1,RRM1-AS1,AC110619.1,AP001029.2,LINC00520 |
| MiRNA | hsa-mir-301b,hsa-mir-206,hsa-mir-31,hsa-mir-383,hsa-mir-375,hsa-mir-503,hsa-mir-210 |
| mRNA | GPD1L,GREM2,HOXC13,STC2,DIO1,FOXD4L1,PAX3 |

**Table 6  The target genes of miRNAs in ceRNA network based on databases.**

| miRNA | LogFC | Adjusted $P$-value | Stage | Target genes |
|---|---|---|---|---|
| hsa-mir-210 | 2.298119 | 1.05E-05 | up | SH3BGRL,ALDH5A1,DENND6A,POU2AF1,SERTM1,VAMP4,FGFRL1,GPD1L, AIFM3,ACVR1,BKCMF1,ISCU,SIN3A,MDGA1 |
| hsa-mir-375 | −3.368088771 | 9.26E-13 | down | RLF,ELAVL4 |
| hsa-mir-206 | −2.44746455 | 0.003311076 | down | WEE1,EIF1AX,FRS2,PGD,FNDC3A,KCNJ2,G6PD,STC2,GPD2,PAX3,ZNF215, LRRC59,TKT,BDNF,BSCL2,GJA1,VAMP2, ANP32B,SFRP1,HSP90B1,KRAS, MATR3,CERS2,RNF138,SMARCB1,UTRN,NUP50 |
| hsa-mir-383 | −2.376630995 | 0.001185218 | down | DIO1,SRSF2,VEGFA,ADSS,IRF1 |
| hsa-mir-31 | 3.292738762 | 0.006118236 | up | TBXA2R,STK40,MZT1,ZNF805,SELE,FOXD4,FZD3,FOXD4L5, SP1,FOXD4L4,ZC3H12C,FOXD4L1,LATS2,SYDE2,PRKCE,RHOBTB1, HIF1AN,RASA1,ARID1A,GTF2E1,NOL9,HOXC13,JAZF1,CCNT1,C19orf12, ECHDC1,NF2,NUMB,ARF1,PPP2R2A,PARP1,ABCB9,YWHAE,KLF13 |
| hsa-mir-503 | 2.371556173 | 1.88E-07 | up | RALGAPB,CREBL2,CCND2,CCND1,GREM2,JARID2,ZNF282,ZNRF2 |
| hsa-mir-301b | 2.864323582 | 8.75E-07 | up | — |

**Notes.**

FC,  fold change.

these key RNAs in ceRNA networks could serve as prognostic biomarkers and their further studies might contribute to LSCC therapy.

## DISCUSSION

Laryngeal squamous cell carcinoma, one of the most common tumors of the head and neck, has been extensively studied to identify its key regulatory genes and molecules. This study comprehensively analyzed the expression and function of aberrant RNAs and established a ceRNA network of LSCC. Sixty-one lncRNAs, seven miRNAs and seven target mRNAs

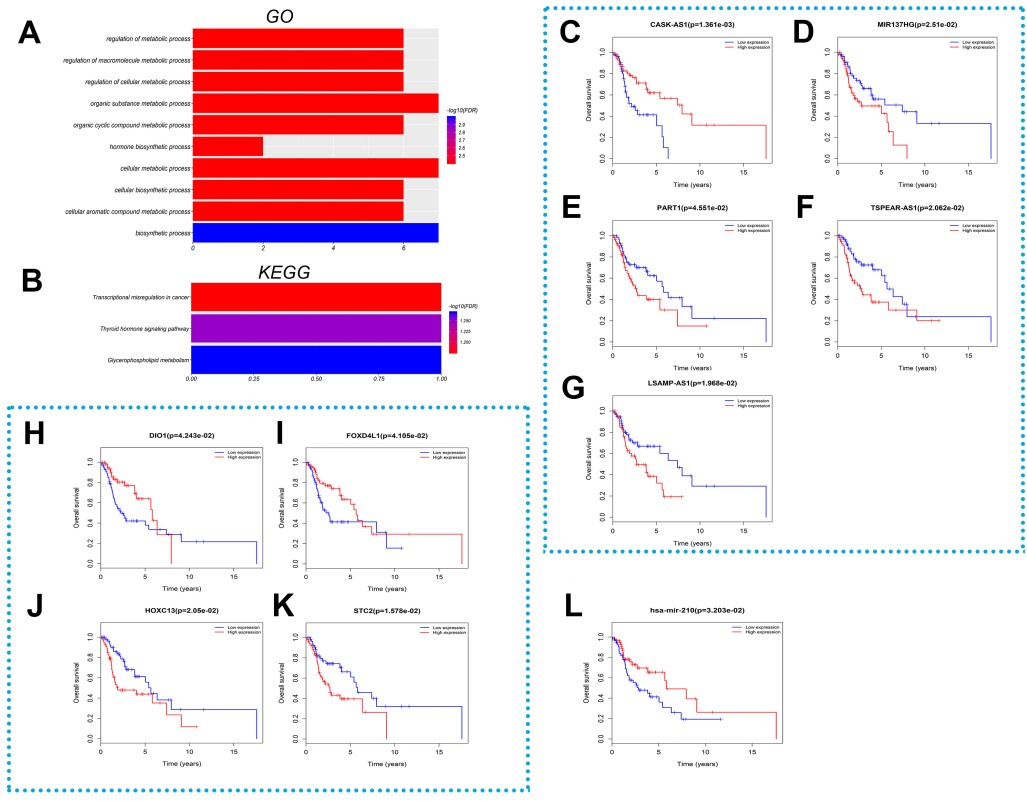

**Figure 4** **Function and survival analysis of RNAs in the ceRNA network.** (A) Significant bioprocesses and (B) pathways representing the functions of mRNA in the ceRNA network. The horizontal axis stands for the number of enriched genes and the color intensity represents corrected *P*-value. Kaplan–Meier curves analysis of differentially expressed (C–G) lncRNAs, (H–K) mRNAs and (L) miRNA for the overall survival in LSCC patients. The horizontal axis represents overall survival time (years), while the vertical axis represents survival function.

were included in the ceRNA network. Through analyzing the overall survival, 5 lncRNAs (TSPEAR-AS, CASK-AS1, MIR137HG, PART1, LSAMP-AS1), miRNAs (has-mir-210), and 4 mRNAs (HOXC13, STC2, DIO1, FOXD4L1) were found to be associated with overall survival in LSCC patients.

Using bioprocess and signaling pathway analyses, the biological functions and potential regulatory mechanisms of aberrantly expressed mRNAs were identified. GO and KEGG enrichment analyses associated these mRNAs with bioprocesses and pathways known to be involved in LSCC, including extracellular matrix organization, calcium signaling and metabolic pathways (Figs. 1D, 1E, and 2). PPI analysis of these altered mRNAs revealed that hub genes including MYH6, ACTN2, TCAP, TNNI2, LCE, SPRR2G, MMP9 and FGA might play crucial roles in LSCC tumorigenesis mechanisms (Fig. 1C).

Recent evidence suggests that extracellular matrix(ECM) organization forms a tissue-specific microenvironment that plays a critical role in tumor progression and metastasis (*Insua-Rodríguez & Oskarsson, 2016*; *Walker, Mojares & Del Río Hernández, 2018*; *Zhou & Lu, 2017*). The ECM is an extremely complex and dynamic molecular

network surrounding tumor cells. As tumor cells proliferate, the surrounding ECM undergoes significant architectural changes through a dynamic interplay between the microenvironment and resident cells (*Grossman et al., 2016*). These changes are consistent with the findings (Fig. 1D) in this work that bioprocess of extracellular matrix organization significantly enriched aberrant mRNAs of LSCC. Interestingly, a type IV collagenase matrix metalloprotease-9 (MMP9), which is identified as one of the most significant gene in PPI analysis, also plays an important role in ECM. MMP9 degrading collagen IV, which is a major component of basement membranes, facilitates tumor cells invasion (*Malik, Lelkes & Cukierman, 2015*). The increasing expression of MMP9 has been reported to be associated with metastasis, and poor prognosis in breast and colon cancer (*Reggiani et al., 2017*; *Yang et al., 2017*). Recent studies have shown that MMP9 increases tumor resistance to anti-PD-1 (*Zhao et al., 2018*). This data indicates that normalization of ECM components and inhibition of MMP9 might be a promising strategy against LSCC. Indeed, previous experiment have predicted this possibility. For example, *Sherman et al. (2014)* used agonist of stromal vitamin D receptor to reprogram ECM by reducing fibrosis and increasing angiogenesis, and hence enhances the efficacy of gemcitabine treatment in pancreatic cancer. Isua-Rodriguez and Oskarsson summarized the target component of ECM in tumor therapy and highlighted the prospect of targeting ECM in breast cancer (*Insua-Rodríguez & Oskarsson, 2016*). Regarding LSCC, the results of the current study suggest that targeting ECM can start with inhibiting MMP9. It is worth noting that matrix metalloproteases (MMPs) are considered as promising targets against cancer in recent decades, but most clinical trials of MMP inhibitor have failed (*Insua-Rodríguez & Oskarsson, 2016*). This is caused by high toxicity and low selectivity (*Coussens, Fingleton & Matrisian, 2002*). Nevertheless, many new generation MMP inhibitor with higher selectivity are currently being developed and tested in cancer (*Ager et al., 2015*; *Kaimal et al., 2013*). Therefore, MMP inhibitors are still promising for the treatment of tumors.

Based on the results of pathway enrichment analysis, it was found that calcium signaling pathway might play an important role in LSCC tumorigenesis. As described in previous studies, calcium signaling is linked to key cell cycle, including early entry into G1 and progression through G1/S and G2/M (*Prevarskaya et al., 2014*; *Roderick & Cook, 2008*). Calcium signaling is also involved in specific oncogene and pro-oncogene pathways by which tumor cells generated resistance to apoptosis (*Di et al., 2015*; *Lange et al., 2016*; *Tang et al., 2016*). In addition, many studies focus on the effect of calcium signaling on tumor microenvironment, which guides tumor invasion and metastasis. For instance, activation of calcium-dependent kinase PNCK, results in macrophage recruitment, angiogenesis, and tumor progression by calcium-dependent NF-κB activation (*Sang et al., 2018*). Calcium signaling increasing HIF1α stability contributes to the progression of hepatocarcinoma (*Li et al., 2015*). It is as if environmental factor, hypoxia, can be transmitted to tumor cells via calcium signaling. Moreover, calcium signaling has been found to directly regulate cancer cells death induced by cytotoxic T lymphocytes and NK cells (*Monteith, Prevarskaya & Roberts-Thomson, 2017*). Targeting calcium signaling is a promising strategy for cancer therapy. Inhibitors of TRPV6, the highly Ca2+ selective ion channel, has now undergone phase I clinical trials in patients with advanced tumors (*Fu et al., 2017*). Targeting calcium

signaling can reactivate tumor suppressor genes silenced by calcium-calmodulin kinase and increase cell death in colon cancer (*Raynal et al., 2016*). Moreover, suppression of Ca2+ channels has be showed to inhibit the proliferation of LSCC cell line *in vitro* (*Yu et al., 2014*). However, to date there are very few studies on calcium signaling in LSCC. In brief, the current results highlighted the importance of ECM and calcium pathways for their potential therapeutic value in LSCC.

LncRNAs, composed of >200 nucleotides (nt) without protein-coding ability, play an important role in epigenetic, transcriptional, post-transcriptional and translational regulation (*Khorkova, Hsiao & Wahlestedt, 2015*). Previous profiling studies suggest that the aberrant expression of lncRNA in LSCC may be a potential mechanism of tumorigenesis and development, and might become biomarkers for diagnosis and prognosis (*Gong et al., 2014*; *Parasramka et al., 2016*; *Rupaimoole et al., 2015*; *Yuan et al., 2014*). As described in Salmena's ceRNA hypothesis, lncRNAs competitively bind to miRNAs via their intramolecular miRNA reaction elements (MREs) to participate in post-transcriptional regulation (*Salmena et al., 2011*). Numerous studies have suggested that ceRNA networks play an important role in the regulation of gene expression in cancers, such as gastric cancer, breast cancer, endometrial cancer, thyroid cancer, and lymphoma (*Guo et al., 2015*; *Wang et al., 2016c*). In this study, the lncRNA-mediated ceRNA regulatory network was analyzed to explore potential novel regulatory mechanisms for lncRNAs in LSCC. Moreover, five lncRNAs (TSPEAR-AS, CASK-AS1, MIR137HG, PART1, LSAMP-AS1) that correlated with the prognosis of patients with LSCC were screened.

As described in previous studies, high lncRNA PART1 expression is associated with poor prognosis and tumor recurrence in non-small cell lung cancer (*Li et al., 2017a*). Elevated PART1 is found in esophageal squamous cell carcinoma and promotes gefitinib resistance by competitively binding to miR-129 to facilitate Bcl-2 expression (*Kang et al., 2018*). Moreover, PART1 upregulated by androgen is reported to promote cell proliferation and inhibit apoptosis via Toll-like receptor pathway in prostate cancer (*Sun et al., 2018*). Interestingly, the larynx is also considered a secondary sexual organ and the androgen receptor is highly upregulated in LSCC (*Fei et al., 2018*). These data indicates that PART1 also play an important role in LSCC. The current results showed that PART1 was negatively correlated to survival rate in LSCC and could be a prognosis biomarker. These results are consistent with findings in other HNSCC, including oral squamous cell carcinoma and tongue squamous cell carcinoma (*Li et al., 2017b*; *Zhang et al., 2019*).

LSAMP-AS1, also called LOC285194 or Tumor Suppressor Candidate 7(TUSC7), was previously defined as a tumor suppressor in esophageal squamous cell carcinoma (ESCC) and colorectal cancer (*Liu et al., 2013*; *Qi et al., 2013*; *Tong et al., 2014*). LSAMP-AS1/miR-224 regulates proliferation, apoptosis, and chemotherapy resistance of ESCC by regulating DESC1/EGFR/AKT pathway (*Chang et al., 2018*). LSAMP-AS1 inhibits proliferation by sponging miR-211 in colorectal cancer (*Xu, Zhang & Zhao, 2017*). Nevertheless, the current study suggested that LSAMP-AS1 might be an up-regulation promotor in LSCC with poor prognosis. A reasonable explanation for this discrepancy might be that LSAMP-AS1 influences the LSCC pathogenesis through other mechanisms. In this work, upregulated LSAMP-AS1 directly inhibited miR-375 in the ceRNA network. MiR-375 is downregulated

and considered as a tumor suppressor in LSCC (*Guo et al., 2016*; *Wang et al., 2016a*). MiR-375 targets IGF1R and affects its downstream AKT signaling pathways which contributes to inhibition of cell progression in LSCC (*Luo et al., 2014*). The results in the current work indicate that downregulation of miR-375 has effects on upregulated LSAMP-AS1. While LSAMP-AS1 may play an important role in LSCC by targeting miR-375. However, these results should be further confirmed *in vivo* and *in vitro*.

The current study identified has-miR-210 related to the diagnosis and prognosis of LSCC. Several existing studies indicate that has-mir-210 may serve as a marker for tumor hypoxia and a prognostic factor in HNSCC (*Gee et al., 2010*; *Huang et al., 2009*). Has-miR-210 is robustly and consistently induced in response to hypoxia. Recent studies showed that has-mir-210 is up-regulated in HNSCC sample and inhibits proliferation of tumor cells (*Zuo et al., 2015*). These data are consistent with the current findings that hsa-miR-210 is up-regulated and positively correlated with prognosis. It has been clearly established that Hypoxia-inducible factor-1 (HIF1), as one of the genes that is induced by hypoxia, directly increases the expression of has-mir-210 (*Gee et al., 2010*; *Huang et al., 2009*; *Merlo et al., 2012*). However, the differential expression of HIF1 between LSCC and normal samples was not found in the current study. This result suggests that other mechanisms are involved in the regulation of has-miR-210 expression in LSCC. In the current ceRNA network, low expression of lncRNA ALDH1L1-AS2 and EMX2OS was involved in up-regulation of has-mir-210. Although the function of these two lncRNA has not been studied, the results indicated that the two lncRNAs regulates the expression of GPD1L by competitively inhibiting has-miR-210. Similarly, hypoxia induced GPD1L can decrease the subunit of HIF1 protein. HIF1 protein mediates hypoxic responses and regulates gene expression involved in angiogenesis, invasion and metabolism (*Kim et al., 2016*). Previous reports showed that GPD1L as a direct target of has-mir-210 is positively associated with prognosis in HNSCC (*Feng et al., 2014*; *Kelly et al., 2011*). The down-regulation of GPD1L by miR-210 occurs *in vivo* and contributes to HIF1 stability, which enhance the metastatic and invasiveness of cancer (*Costales et al., 2017*; *Gee et al., 2014*; *Kelly et al., 2011*). Therefore, ALDH1L1-AS2 or EMX2OS targeting has-miR-210/GPD1L may regulate LSCC development by HIF1.

In this study, we used bioinformatics to analyze the expression and function of aberrantly expressed lncRNA, miRNA, and mRNA in LSCC and establish a ceRNA network to predict the regulation mechanism of lncRNA. In a previous report, *Feng et al. (2016)* used a similar method to establish ceRNA network in LSCC. *Feng et al. (2016)* analyzed differentially expressed lncRNA and mRNA via Microarray assay. Some results of Feng's study are similar to ours, that the extracellular matrix, metabolic pathway, and hypoxia-induced HIF1 are important in LSCC. However, Feng et al. did not perform a differential analysis of miRNAs. This may lead to the fact that the miRNAs in their ceRNA networks are not abnormally expressed in LSCC. In addition, *Feng et al. (2016)* did not analyze the prognostic significance of the screened RNA. In our study, expression data we used comes from high-throughput sequencing and contain a larger sample size. The aberrantly expressed lncRNAs, miRNAs, and mRNAs in LSCC were utilized to construct the ceRNA network and the survival analysis of them identified their prognostic significance. However, some limitations of the

current study should be acknowledged. The function of lncRNA is complex. This study only analyzed one mechanism of lncRNA competitively combining miRNA. Moreover, the database, used to predict interactions between these RNAs, is outdated. This may had caused the omission of some important information. Contradictory results may occur between different bioinformatics analyses because of different platforms, different parameter settings and different correction methods. Therefore, further experimental or clinical studies are needed to validate these results.

## CONCLUSIONS

The current study identified differentially expressed lncRNA, miRNA and mRNA via RNA-Seq data of large-scale samples from TCGA. GO and KEGG enrichment analysis identified the regulatory roles of altered RNAs but the precise regulatory mechanisms need further study. The construction of the ceRNA network broadens the understanding of mechanisms by which lncRNAs are involved in tumorigenesis. In this network, five lncRNAs (TSPEAR-AS, CASK-AS1, MIR137HG, PART1, LSAMP-AS1), miRNA (has-mir-210), and four mRNAs (HOXC13, STC2, DIO1, FOXD4L1) were found to be associated with overall survival of LSCC patients. It is noteworthy that the results in this study were predicted via bioinformatics and therefore should be verified through *in vivo* and *in vitro* experiments.

## ACKNOWLEDGEMENTS

We are sincerely grateful to the BIOWOLF for providing guidance in bioinformatics. We also acknowledge the editorial assistance provided by FREESCIENCE.

### Funding

The authors received no funding for this work.

### Competing Interests

The authors declare there are no competing interests.

### Author Contributions

- Yuehui Liu conceived and designed the experiments, performed the experiments, analyzed the data, contributed reagents/materials/analysis tools, authored or reviewed drafts of the paper.
- Fan Ye conceived and designed the experiments, performed the experiments, analyzed the data, prepared figures and/or tables, approved the final draft.

### Data Availability

Data is available at TCGA (https://portal.gdc.cancer.gov/repository) under ID:TCGA-DQ-5629, TCGA-CV-7437, TCGA-CN-6992,TCGA-CN-6021TCGA-CN-6022, TCGA-CV-7440, TCGA-BA-6870, TCGA-CV-7410, TCGA-HD-7229, TCGA-CN-6997, TCGA-CN-4727, TCGA-CN-6023TCGA-CN-A641, TCGA-BA-4076, TCGA-CN-4723, TCGA-CV-7433, TCGA-CV-7177, TCGA-CN-5355, TCGA-CN-5356, TCGA-CV-7415, TCGA-CN-4735, TCGA-CN-6988, TCGA-CN-5363, TCGA-CR-7398, TCGA-BA-6868, TCGA-CV-5441, TCGA-CN-6012, TCGA-CV-5444, TCGA-D6-A74Q, TCGA-CN-A6V3, TCGA-UF-A71D, TCGA-H7-A6C5, TCGA-CN-6989, TCGA-CV-5443, TCGA-D6-6826, TCGA-CV-7422, TCGA-UF-A718, TCGA-CV-5440, TCGA-D6-A6EK, TCGA-CR-7370, TCGA-CV-7245, TCGA-UF-A7JF, TCGA-CN-A497, TCGA-CR-6474, TCGA-CN-A49B, TCGA-TN-A7HJ, TCGA-CR-7402, TCGA-CR-7371, TCGA-CV-5430, TCGA-CN-A63U, TCGA-CV-5432, TCGA-QK-AA3J, TCGA-CV-7250, TCGA-CV-5431, TCGA-CV-7248, TCGA-CV-7261, TCGA-CN-A63W, TCGA-CN-4722, TCGA-CV-7418, TCGA-CN-A63T, TCGA-BA-6869, TCGA-CV-A6K1, TCGA-CR-7389, TCGA-CV-7430, TCGA-CR-7399, TCGA-F7-7848, TCGA-BA-A6DI, TCGA-KU-A66S, TCGA-D6-6517, TCGA-D6-A6ES, TCGA-CV-7089, TCGA-CV-A45Z, TCGA-F7-A623, TCGA-CV-A461, TCGA-CV-7424, TCGA-F7-A50I, TCGA-UF-A7JK, TCGA-CV-6935, TCGA-CN-4739, TCGA-BA-A6DA, TCGA-QK-A8Z8, TCGA-CV-7421, TCGA-CV-A45Y, TCGA-UF-A7JJ, TCGA-CV-A460, TCGA-CV-7242, TCGA-CN-6010, TCGA-F7-A622, TCGA-D6-8568, TCGA-UF-A7J9, TCGA-D6-A6EQ, TCGA-CR-7364, TCGA-CR-7388, TCGA-BA-5555, TCGA-CV-6962, TCGA-QK-A8ZB, TCGA-CV-A45W, TCGA-T3-A92M, TCGA-F7-8298, TCGA-CN-5360, TCGA-CR-7374, TCGA-D6-6824, TCGA-BB-4217, TCGA-CN-4738, TCGA-CV-7247, TCGA-CV-5435, TCGA-BA-4078, TCGA-CV-5978, TCGA-CN-5361, TCGA-CV-5434, TCGA-UF-A7JH, and TCGA-CV-7101.

## Supplemental Information

Supplemental information for this article can be found online at http://dx.doi.org/10.7717/peerj.7380#supplemental-information.

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
