# Peer review of "Construction and integrated analysis of crosstalking ceRNAs networks in laryngeal squamous cell carcinoma"

_PeerJ, doi:10.7717/peerj.7380_

## Round 0.1 · original submission · Major Revisions

Manuscript "Construction and integrated analysis of crosstalking ceRNAs networks in laryngeal squamous cell carcinoma" which you submitted to PeerJ, has been reviewed. The reviewers have recommended publication pending major revisions. Therefore, I invite you to respond to the reviewers' comments at the bottom of this letter and revise your manuscript accordingly.

Reviewer 1 ·

Basic reporting

no comments

Experimental design

no comments

Validity of the findings

I appreciate the chance to review this paper, although there is great effort in pre-paring this manuscript, the main drawback is it do not add any information we have already known.
1. The discussion section is too simple and superficial, the authors must revise it in deep.
2. In the screened RNA, cytological validation or molecular biology validation in tissues is recommended.

·

Basic reporting

In this paper, Liu et al. have investigated the functions of lncRNA as ceRNA in LSCC from the TCGA. The results suggested that the potential regulatory roles of different RNAs might be the mechanisms of LSCC tumorigenesis. Overall, the topic is interesting and the data is robust, statistically sound.

Experimental design

no comment

Validity of the findings

no comment

Additional comments

1. The standard of English is poor and unacceptable, partly so that no sense can be derived from sentences as they stand. There are a lot of grammatical mistakes, omitted words, and phrases not commonly used in English.
2.The authors write “a systematic understanding of how ceRNA network contributes to tumorigenesis and progression of LSCC is still lacking” (lines 83-84). Actually, several studies have already reported the comprehensive analysis of the coding and non-coding RNA in LSCC from tissues before. Other literatures have also reported ceRNA network in recurrence LSCC from TCGA. The paper didn’t have mentioned these relevant prior literatures at all. The novelty of this paper remains to be proved.
3. The section of Abstract needs to be polished. The results are too general and the prognostic significance didn’t have been mentioned in conclusion.
4. The basic clinical information should be provided in detail, such as age, TNM stage, vital status, subdivision.

Reviewer 3 ·

Basic reporting

- Punctuation, spaces issues and capital words in the middle of the sentences throughout the manuscript need to be corrected.
- Line 380, figure number is missing
- Acknowledgements were not added

Experimental design

Competitive endogenous RNA network has been used with the same approach in many articles.
- For figures 4B, 4C and 4D, how did the authors determine the cutoff for low and high expression.

Validity of the findings

The paper would be more relevance by performing an experimental validation to support some of the computational results.The authors need to validate the interaction of the suggested lncRNA-miRNA-mRNA network.

Additional comments

In this manuscript, the authors analyzed expression profiles of lncRNA, miRNA ad mRNA in laryngeal squamous cell carcinoma (LSCC) from the TCGA database. Gene Ontology (GO) as well as Kyoto Encyclopedia of Genes and Genomes (KEGG) pathway were analyzed and a ceRNA network was constructed. The authors identified potential regulatory lncRNA, miRNA and mRNA that might be one of the mechanisms of LSCC tumorigenesis. Overall, the work is interesting.

---

## Round 0.2 · accepted · Accept

I am pleased to inform you that your manuscript "Construction and integrated analysis of crosstalking ceRNAs networks in laryngeal squamous cell carcinoma" has been accepted for publication in PeerJ.

·

Basic reporting

no comment

Experimental design

no comment

Validity of the findings

no comment

Additional comments

The authors have well revised the manuscript. I think this version is worth to be accepted.

Reviewer 3 ·

Basic reporting

no comment

Experimental design

no comment

Validity of the findings

no comment

Additional comments

The authors have responded to all concern previously raised by the reviewer and have adequately addressed them.